# The potential health impact of restricting less-healthy food and beverage advertising on UK television between 05.30 and 21.00 hours: A modelling study

Oliver T. Mytton[1]*, Emma Boyland[2], Jean Adams[1], Brendan Collins[3], Martin O'Connell[4], Simon J. Russell[5], Kate Smith[4], Rebekah Stroud[4], Russell M. Viner[5], Linda J. Cobiac[6]

1 Centre for Diet and Activity Research, MRC Epidemiology Unit, University of Cambridge, Cambridge, United Kingdom, 2 Department of Psychological Sciences, University of Liverpool, Liverpool, United Kingdom, 3 Department of Public Health and Policy, University of Liverpool, Liverpool, United Kingdom, 4 Institute for Fiscal Studies, London, United Kingdom, 5 Great Ormond Street Institute of Child Health, University College London, London, United Kingdom, 6 Centre on Population Approaches for Non-Communicable Disease Prevention, Nuffield Department of Population Health, University of Oxford, Oxford, United Kingdom

* otm21@medschl.cam.ac.uk

## Abstract

### Background

Restrictions on the advertising of less-healthy foods and beverages is seen as one measure to tackle childhood obesity and is under active consideration by the UK government. Whilst evidence increasingly links this advertising to excess calorie intake, understanding of the potential impact of advertising restrictions on population health is limited.

### Methods and findings

We used a proportional multi-state life table model to estimate the health impact of prohibiting the advertising of food and beverages high in fat, sugar, and salt (HFSS) from 05.30 hours to 21.00 hours (5:30 AM to 9:00 PM) on television in the UK. We used the following data to parameterise the model: children's exposure to HFSS advertising from AC Nielsen and Broadcasters' Audience Research Board (2015); effect of less-healthy food advertising on acute caloric intake in children from a published meta-analysis; population numbers and all-cause mortality rates from the Human Mortality Database for the UK (2015); body mass index distribution from the Health Survey for England (2016); disability weights for estimating disability-adjusted life years (DALYs) from the Global Burden of Disease Study; and healthcare costs from NHS England programme budgeting data. The main outcome measures were change in the percentage of the children (aged 5–17 years) with obesity defined using the International Obesity Task Force cut-points, and change in health status (DALYs). Monte Carlo analyses was used to estimate 95% uncertainty intervals (UIs). We estimate that if all HFSS advertising between 05.30 hours and 21.00 hours was withdrawn, UK children (n = 13,729,000), would see on average 1.5 fewer HFSS adverts per day and decrease

**Data Availability Statement:** We used the following data sources: AC Nielsen and Broadcasters' Audience Research Board data on

children's exposure to HFSS advertising (2015) (data is not freely available, it is owned by AC Nielsen and whilst we had access to the full dataset for our own analysis, the terms of use prohibit us sharing the dataset; https://www.barb.co.uk/); Kantar Worldpanel dataset for the UK for information on the nutrient content of foods (data is not freely available, it is owned by Kantar and whilst we had access to the full dataset for our own analysis, the terms of use prohibit us sharing the dataset; https://www.kantarworldpanel.com/en) published meta-analysis quantifying the effect of less-healthy food advertising on acute caloric intake in children (freely available and published by Obesity Reviews, it is open access); population numbers and all-cause mortality rates from the Human Mortality Database for the UK (2015) (freely available at www.mortality.org); body mass index distribution from the Health Survey for England (2016) (freely available from NHS Digital (summary reports; https://digital.nhs.uk/data-and-information/publications/statistical/health-survey-for-england/health-survey-for-england-2016) and UK Data Archive (raw data sets; https://beta.ukdataservice.ac.uk/datacatalogue/series/series?id=2000021)); disability weights for estimating Disability Adjusted Life Years (DALYs) were taken from the Global Burden of Disease Study (freely available, http://ghdx.healthdata.org/gbd-results-tool); healthcare costs were taken from NHS England programme budgeting data (freely available using the estimated reported by Briggs et al in PLOS ONE, open access publication).

**Funding:** This study did not receive any specific funding. OM is an NIHR Clinical Lecturer. MO is funded under the British Academy Postdoctoral Fellowship scheme. JA has received funding from the Centre for Diet and Activity Research (CEDAR), a UKCRC Public Health Research Centre of Excellence (RES-590-28-0002); within this funding from the British Heart Foundation, Department of Health, Economic and Social Research Council, Medical Research Council and the Wellcome Trust, under the auspices of the UK Clinical Research Collaboration, is gratefully acknowledged. The funders had no role in study design, data collection and analysis, decision to publish, or preparation of the manuscript.

**Competing interests:** I have read the journal's policy and the authors of this manuscript have the following competing interests: RV MO RS KS report a grant for the Obesity Policy Research Unit, funded through the National Institute for Health Research (NIHR) to inform the work of the Department for Health and Social Care; OM was an

caloric intake by 9.1 kcal (95% UI 0.5–17.7 kcal), which would reduce the number of children (aged 5–17 years) with obesity by 4.6% (95% UI 1.4%–9.5%) and with overweight (including obesity) by 3.6% (95% UI 1.1%–7.4%) This is equivalent to 40,000 (95% UI 12,000–81,000) fewer UK children with obesity, and 120,000 (95% UI 34,000–240,000) fewer with overweight. For children alive in 2015 ($n = 13,729,000$), this would avert 240,000 (95% UI 65,000–530,000) DALYs across their lifetime (i.e., followed from 2015 through to death), and result in a health-related net monetary benefit of £7.4 billion (95% UI £2.0 billion–£16 billion) to society. Under a scenario where all HFSS advertising is displaced to after 21.00 hours, rather than withdrawn, we estimate that the benefits would be reduced by around two-thirds. This is a modelling study and subject to uncertainty; we cannot fully and accurately account for all of the factors that would affect the impact of this policy if implemented. Whilst randomised trials show that children exposed to less-healthy food advertising consume more calories, there is uncertainty about the nature of the dose–response relationship between HFSS advertising and calorie intake.

## Conclusions

Our results show that HFSS television advertising restrictions between 05.30 hours and 21.00 hours in the UK could make a meaningful contribution to reducing childhood obesity. We estimate that the impact on childhood obesity of this policy may be reduced by around two-thirds if adverts are displaced to after 21.00 hours rather than being withdrawn.

## Author summary

### Why was this study done?

- Watching unhealthy food advertising increases the calories children eat.

- While greater calorie intake increases the likelihood of a child gaining excess weight, little is known about the impact of advertising on childhood obesity and overweight.

- Restrictions on television advertising of unhealthy food are actively being considered in the UK to help prevent childhood obesity; the impact of these proposed restrictions is unknown.

### What did the researchers do and find?

- We used computer modelling to estimate the health impact of prohibiting the advertising of food and beverages high in fat, sugar, and salt (HFSS) from 05.30 hours to 21.00 hours on television in the UK. Using 2015 data, we estimate that if all HFSS advertising between 05.30 hours and 21.00 hours was withdrawn, children in the UK would see, on average, 1.5 fewer HFSS adverts per day, which would reduce the number of children (aged 5–17 years) with obesity by 4.6% (95% uncertainty interval [UI] 1.4%–9.5%) and with overweight (including obesity) by 3.6% (95% UI 1.1%–7.4%).

advisor to the Health Select Committee during their inquiry into childhood obesity (2017-18).

**Abbreviations:** BARB, Broadcasters' Audience Research Board; DALY, disability-adjusted life year; HFSS, high in fat, sugar, and salt; IOTF, International Obesity Task Force; SES, socioeconomic status; UI, uncertainty interval.

- We estimate that if all HFSS advertising is displaced to after 21.00 hours, rather than being withdrawn, the health benefits would be reduced by around two-thirds.

## What do these findings mean?

- Measures that have the potential to reduce exposure to less-healthy food advertising on television, such as restricting HFSS advertising between 05.30 hours and 21.00 hours, could make a meaningful contribution to reducing childhood obesity.

- The impact of this policy is likely to be reduced if adverts are displaced to after 21.00 hours or to other media.

- This is a modelling study, and we cannot fully account accurately for all factors that would affect the impact of this policy if it was implemented.

## Introduction

Childhood obesity is a global problem, with few signs of progress [1]. Overweight and obesity directly affects children's health and well-being [2,3], increases their likelihood of attendance at primary care [4], and may adversely affect educational achievement [5]. Children with obesity are 5 times more likely to have obesity as an adult [6], and are at increased risk of premature death and of developing a range of diseases in adult life (e.g., cardio-metabolic disease, some cancers) [7,8].

In England, 1 in 5 children aged 4–5 years and 1 in 3 children aged 10–11 years have overweight or obesity [9]. The UK government has announced ambitious plans to halve childhood obesity by 2030 [9]. As part of its plan to reduce childhood obesity, it is consulting on a 9:00 PM watershed for advertising for foods or beverages ('products') high in fat, sugar, and salt (HFSS), as defined the by the Food Standards Agency nutrient profiling model, on television [9,10]. This entails restricting television advertising for most HFSS products between 05.30 hours (5:30 AM) and 21.00 hours (9:00 PM). The proposed measure has received strong support from the Chief Medical Officer [11], the Health Select Committee [12], and health bodies [13].

The proposal represents a tightening of the current regulations in the UK, introduced in 2007, which prohibit the advertising of HFSS products in or around programmes of particular appeal to children aged 4–15 years and on dedicated children's channels. While an official report suggested that these restrictions did reduce children's exposure to advertising [14], independent research suggested that less-healthy food advertising switched away from dedicated children's programmes to family programmes that are watched by a large number of children [15,16], such that children's exposure was largely unchanged. Recent work has shown that children still see substantial HFSS advertising on television, despite good adherence to the current regulation [15], with much of this exposure happening before 21.00 hours [17].

Since 2007, evidence has continued to accumulate showing that food and beverage advertising contributes to increased calorie intake in children [18,19]. Meta-analyses of short-term trials have consistently shown that children exposed to television advertisements for less-healthy food consume more food in the immediate period after watching such advertisements [18–

20]. As children do not appear to compensate by adequately reducing consumption at subsequent meals [21], the repeated exposure to television advertisements for less-healthy foods is likely to lead to repeated over-consumption and excess weight gain. In addition, evidence suggests that exposure to advertising for less-healthy foods influences food preferences and purchasing patterns [22,23]. This is likely to encourage consumption of a diet high in HFSS foods, or a Western dietary pattern, which is associated with obesity and ill-health [24–26].

The World Health Organization recommends restricting television advertising to children for products high in saturated fat, trans fatty acids, free sugar, or salt [27]. While many countries have taken action to limit less-healthy food advertising to children, no country to our knowledge has yet implemented a 9:00 PM watershed [28,29]. A cross-country comparison found that those countries that had statutory regulations experienced a smaller increase in household expenditure on less-healthy food and drink products over time [28]. A recent study estimated that a 9:30 PM watershed in Australia would reduce mean BMI by 0.35 kg/m$^2$ amongst children, be cost saving, and reduce inequalities in childhood obesity. Potential differences in television viewing patterns and advertising practices between the UK and Australia mean those results are not readily transferable to the UK [30].

Understanding the potential size of health impacts, consequent cost savings, and economic benefits from restricting HFSS advertising between 05.30 hours and 21.00 hours could help inform the decision about whether and how to introduce the policy in the UK and elsewhere. Sections of the television and advertising industry have questioned the link between advertising and childhood obesity and have expressed concerns about loss of revenue [31,32]. This paper aims to quantify the potential health and consequent monetary benefit of restricting HFSS advertising on television between 05.30 hours and 21.00 hours in the UK.

## Methods

We used data on children's exposure to HFSS advertising on television by time of day to estimate the impact of restricting HFSS advertising between 05.30 hours and 21.00 hours on the amount of HFSS advertising seen by children [17]. We used the most recent meta-analysis of the effect of watching less-healthy food advertising on calorie consumption from experimental studies to estimate the effect of a reduction in HFSS advertising on children's calorie consumption [19]. We used the validated Hall equations to describe the effect of change in calorie consumption on body weight, and then used the PRIMEtime multi-state life table model [33] to estimate the effects of BMI on children's quality of life and their subsequent health in adult life. An overview of our approach, which draws on the methods used by others [30], is shown in Fig 1.

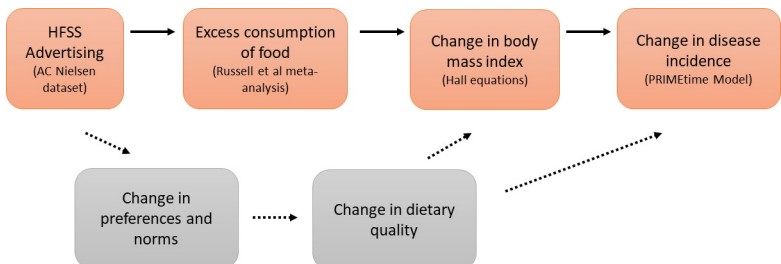

**Fig 1. Overview of the modelling approach used and the pathways through which advertising affects health.**
Orange boxes and bold arrows show the modelled pathway, with key data sources or methods shown in parentheses; dashed arrows and grey boxes highlight a second pathway through which advertising is likely to affect health and which has not been modelled. HFSS, high in fat, sugar, and salt.

The original analytic plan and changes to that plan are provided in S1 Text and S2 Text.

Fig 1 shows the 2 main pathways through which HFSS advertising is thought to affect dietary consumption and health [22,23]. We have modelled the first of these 2 pathways (shown in orange). The second pathway (shown in grey) results from the effect of HFSS advertising in changing food choices and shifting food consumption towards HFSS foods. Existing trial and epidemiological data did not allow us to quantify this second pathway; consequently, it was not modelled.

## Population and scope

Our analysis is focused on children in the UK, reflecting both the current policy focus in the UK and the evidence base, which shows an effect of less-healthy food advertising on excess food consumption only in children [18]. The model was parameterised using data for 2015 (or 2016 where 2015 data were not available). Reflecting current studies, we have assumed that only children aged 5–14 years were affected by exposure to HFSS advertising. The upper limit (14 years) reflects the upper age of children included in studies showing that exposure to less-healthy television food advertising promotes excess calorie consumption [19]. The lower limit (5 years) reflects the lower age limit for which the Hall equations are validated [34] We have assumed that reduction in childhood obesity tracks into adult life, and thus do consider health benefits in later life.

This focus, considering effects only on children age 5–14 years and considering only 1 pathway between advertising and health (i.e., that concerned with the acute effect of advertising on calorie consumption), is likely to underestimate the impact of HFSS television advertising on dietary behaviour and health.

## Advertising exposure

We used data from AC Nielsen on all adverts for food and drink broadcast on television between 1 January and 31 December 2015. The dataset included details on what products and/or brands were advertised, the time of the advert, the channel on which the advert was shown, and what programmes ran either side of the advert. The data also contained information on 'child impacts' for each advert, provided by the Broadcasters' Audience Research Board (BARB). Impacts are a widely used industry measure. One impact represents 1 person viewing 1 advert. Therefore, 2 impacts can represent 2 different people viewing the same advert once, or 1 person viewing the same advert twice.

In this dataset, children were defined as those aged 4–15 years. BARB calculated the number of child impacts by using a panel of 5,100 homes. The panel is reported to be representative of the UK in terms of demography, geography, ethnicity, and TV platform, although BARB does not provide a specific breakdown [35]. Each household was given a remote control with a button on it for each member of the household; each individual pressed their button each time they entered or left a room in which a television was on. Weightings were used to estimate the total number of child impacts for the whole of the UK. We assumed the average number of impacts seen by a child in this dataset (defined as aged 4–15 years) was the average number of impacts seen by children aged 5–14 years, i.e., that impacts were spread widely across the population, given that 96% of children aged 5–15 years are reported to watch television in the UK [35].

## Classification of adverts

We combined the advertising data with information on the nutrient content of products from Kantar Worldpanel for 2015. Foods and non-alcoholic beverages were classified as being HFSS

on the basis of the UK Food Standards Agency nutrient profiling model, which assigns a score to a food or beverage product (based on content per 100 g of the product) for energy; saturated fat; sugar; sodium; fruit, vegetable, and nuts; fibre; and protein [10]. Based on the algorithm [10], each food or beverage was given a single 'nutrient profiling score'; foods and beverages that had a score at or above the threshold were classified as HFSS and were assumed to be subject to restrictions. We refer to products that are at or above the threshold as HFSS, and those that are below the threshold as non-HFSS.

Some adverts were for a brand or product range that contains numerous individual products, rather than a single product. For these adverts, we matched each advert to the set of products that it directly promoted. When all of those products were HFSS, we classified the advert as an HFSS advert (and thus subject to restrictions); when all of the products were non-HFSS, we classified the advert as a non-HFSS advert (and thus not subject to restrictions). A proportion of food and beverage adverts (13% of impacts) were for brands or product sets that included both HFSS and non-HFSS products. For these, we apportioned child impacts to the HFSS/non-HFSS categories in proportion to the market share of products that were HFSS/non-HFSS within the relevant product set. We assumed that adverts for restaurants and bars (the majority of which were for takeaway outlets) included HFSS foods. We assumed that adverts for supermarkets did not include HFSS foods, and thus would not be subject to the restrictions. Only a minority (17.8%) of supermarket adverts on television are for 'non-core' foods (i.e., less-healthy foods) [36], and we assumed that supermarket advertising containing HFSS products would be modified to remove these products.

The number of child advertising impacts for HFSS products by time of day is shown in Fig 2.

## Scenarios

We considered 2 scenarios for how HFSS television advertising would change in response to restrictions: In scenario A, all existing HFSS adverts between 05.30 hours and 21.00 hours are withdrawn. In scenario B, all existing HFSS adverts previously shown between 05.30 hours and 21.00 hours are moved to between 21.00 hours and 05.30 hours.

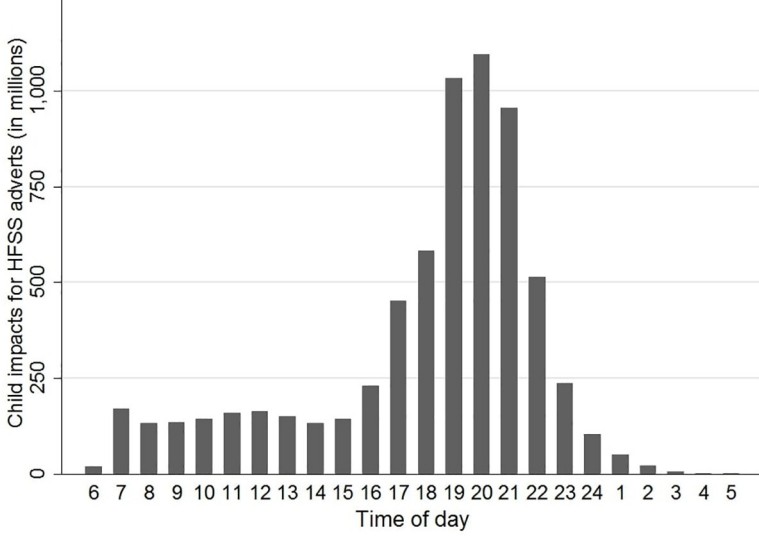

**Fig 2. Children's exposure to HFSS television advertising in the UK by time of day in 2015.** Children defined as ages 4–15 years. An impact is 1 person-viewing of an advert; therefore, 2 impacts could refer to 2 different people viewing an advert once or 1 person viewing an advert twice. HFSS defined by the Food Standards Agency–Ofcom model of 2007 [10]. HFSS, high in fat, sugar, and salt.

To model the shift in advertising in scenario B, we defined 4 time slots: (i) non-prime time before 21.00 hours: 05.30–17.00 hours; (ii) non-prime time after 21.00 hours: 23.00–05.30 hours; (iii) prime time before 21.00 hours: 17.00–21.00 hours; and (iv) prime time after 21.00 hours: 21.00–23.00 hours. We assumed that adverts that were banned during non-prime time before 21.00 hours would shift to a non-prime time slot after 21.00 hours on the same channel. Similarly, we assumed that adverts that were banned during prime time before 21.00 hours would shift to a prime time slot after 21.00 hours on the same channel. We assumed that the number of child impacts for each displaced advert was equivalent to the average number of child impacts for food adverts shown on that channel in that time slot, i.e., if an advert on channel X was shifted from 18.00 (prime time before 21.00 hours) to the prime time slot after 21.00 hours (i.e., 21.00–23.00 hours), the number of child impacts it received would be the average number of child impacts for a food advert shown on channel X between 21.00 hours and 23.00 hours before the intervention. The modelled effect of these scenarios is shown in Fig 3.

## Effect of advertising exposure on energy consumption

We converted the reduction in number of adverts seen to a reduction in the number of minutes of exposure to HFSS advertising. To do this we assumed a mean duration of 25.9 seconds (standard deviation 11.9 seconds) for HFSS adverts, based on reported advert duration in the AC Nielsen dataset. We then converted the reduction in advertising exposure (measured in minutes) to a reduction in calories consumed. To do this, we assumed a linear relationship between time exposed to HFSS advertising and excess calorie consumption following the approach used by others [30]. We assumed that each minute of HFSS advertising seen per child per day equated to an additional 14.2 kcal (95% CI 0.7 to 27.7) consumed per child per day, based on a re-analysis of a recent meta-analysis to express the estimate in kcal/minute [19].

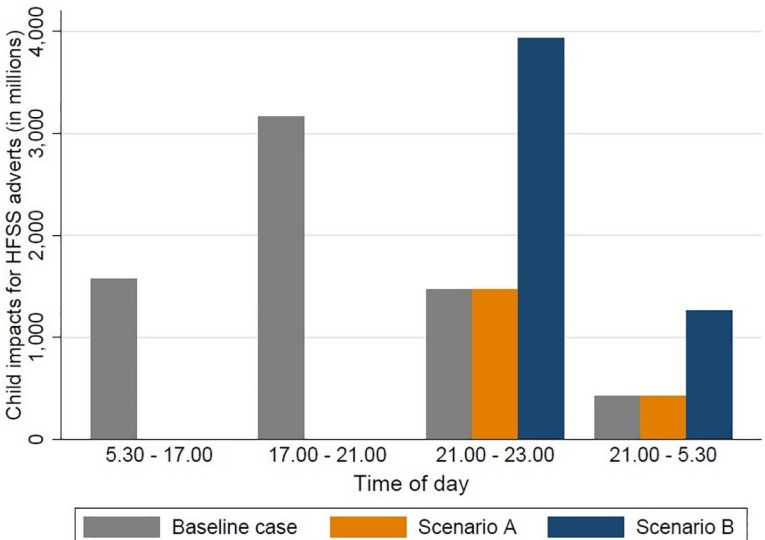

**Fig 3. The modelled effect of the 2 scenarios on children's exposure to HFSS advertising by time of day compared to baseline.** Scenario A: all advertising between 05.30 hours and 21.00 hours is withdrawn. Scenario B: all advertising between 05.30 hours and 21.00 hours is displaced to between 21.00 hours and 05.30 hours. HFSS, high in fat, sugar, and salt.

## Effect of calorie consumption on body mass index

We used the validated Hall equations to estimate the effect of the average change in calorie consumption on the average body weight of boys and girls [34]. We assumed BMI was log-normally distributed. We then assumed that a reduction in mean BMI resulted in a shift of the BMI distribution to the left. As the curve is log-normally distributed, a shift to the left has the effect of reducing the rightwards skew of the current BMI distribution and making the distribution a closer approximation to the normal distribution, i.e., it implicitly assumes a greater calorie reduction and BMI reduction in children who are already overweight. We assumed the shift in the BMI curve that was observed at age 15 years persisted into later life, reflecting the observational data showing that childhood weight status tracks into adult life [37,38]. Children aged between 0 and 4 years at the start of the simulation did not receive an effect until reaching the age of 5 years, reflecting the lower age limit for which the Hall equations have been validated [34]. Baseline BMI distributions and average height, by age and sex, were taken from the Health Survey for England (2016) [39].

## Obesity and overweight

We report the change in the number of children, aged 5–17 years, with obesity and with overweight in the UK. Obesity (BMI for age: >98.9th centile for boy and >98.6th centile for girls based on the WHO growth charts) and overweight (>90.5th centile and >89.3th centile) were based on the International Obesity Task Force (IOTF) definitions [40]. In addition, we also report the number of children with obesity and overweight using UK90 growth charts, using the 95th and 85th centile of BMI for age, which are widely used in the UK.

## Differences by social economic status

We estimated the impact of the restrictions (scenario A) by social grade (see Table 1), making allowance for reported differences in body mass index and television viewing by social grade. We combined social grades A and B (grade AB) and social grades D and E (grade DE).

Based on a survey of television viewing habits in 1 UK region in 2009, we assumed children in social grade DE watched 2.11 times more television than children in social grade AB, and consequently 2.11 times more unhealthy food advertisements. We assumed children in social grade C watched 1.45 times more than children in social grade AB [41].

Weighting by population size (based on 2011 census data with social grade of the head of the household) [42] and applying these values to the 2015 dataset on HFSS advertising on television, for scenario A this equated to 0.97, 1.42, and 2.06 fewer HFSS adverts seen per day by children in social grade AB, grade C, and grade DE, respectively.

Differences in BMI by social grade were taken from the Millennium Cohort Study, a cohort of 19,000 children born in 2000–2001 in the UK, using data from children aged 7, 11, and 14

**Table 1. Definition of social grade in the UK.**

| Social grade | Occupation of head of household |
|---|---|
| A | Higher managerial, administrative, or professional |
| B | Intermediate managerial, administrative, or professional |
| C | Supervisory or clerical and junior managerial, administrative, or professional; skilled manual workers |
| D | Semi-skilled and unskilled manual workers |
| E | State pensioners, casual and lowest grade workers, unemployed with state benefits only |

Source: National Readership Survey: http://www.nrs.co.uk/nrs-print/lifestyle-and-classification-data/social-grade/.

years (i.e., born in 2007–2008, 2011–2012, and 2014–2015), equating the 5-point scale of social class reported in the study to social grade [43]. Weighting by population size, we used the absolute differences in BMI observed in the Millennium Cohort Study by age (see Table A in S1 Data). We used linear interpolation and extrapolation to estimate differences in BMI for age groups for which data were not directly reported. The values we used, including our estimates for age groups with missing data, are shown in S1 Data.

## Health impact modelling

We simulated the long-term health and cost impacts of advertising restrictions in UK children and adolescents using the Excel-based PRIMEtime model [33]. Full details of the model are available elsewhere [33]. Briefly, the model design is based on proportional multi-state life table methods, which are an efficient way of simulating population impacts of changes in risk factors on multiple diseases simultaneously [44]. It was first developed by Cobiac et al. [33] for evaluating health impacts of dietary change scenarios. Briggs et al. then added components allowing the simulation of health and social care costs [45].

For these analyses, we made the following changes to the PRIMEtime model: (i) updated the population numbers and all-cause mortality rates to a baseline year of 2015, using UK data from the Human Mortality Database [46]; (ii) expanded the model to include simulation of children and adolescents; and (iii) redesigned the model structure to allow probabilistic sensitivity analyses using macros rather than an add-in program that relies on ongoing third-party updating.

We simulated a closed cohort of the UK population aged 0–17 years in 2015 ($n = 13,729,000$) and followed them through their lifetime. We simulated the effect of changes in BMI distribution arising from the scenarios on incidence, prevalence, and mortality of all obesity-related diseases, over time, until the simulated population had all died. From the changing disease rates, we determined the impact on total years of healthy life that would be lived in the future.

We report disability-adjusted life years (DALYs), which were estimated by adjusting the healthy years of life lived for time spent in ill-health in adulthood using age- and sex-specific disease disability rates derived from UK Global Burden of Disease Study data. We present both the change in total DALYs undiscounted and an estimate of the discounted DALYs following the approach used by the UK Treasury [47]. This assumes a discount rate of 1.5% for the first 30 years, 1.3% for the next 45 years, and 1.1% thereafter.

The reduction in incidence of obesity-related diseases was estimated using relative risks from meta-analyses of the dose–response relationship between BMI and disease. The obesity-related diseases explicitly modelled were coronary heart disease, stroke, type 2 diabetes, breast cancer (in women), colorectal cancer, pancreatic cancer, kidney cancer, and liver cancer. Diseases were included if there was evidence of an association, e.g., from meta-analyses of prospective cohort studies (or randomised controlled trials); the association was quantified; and the disease makes a substantial contribution to mortality (greater than 500 deaths in the UK in 2006) [48]. Given uncertainty about the immediacy of the effect of change in BMI on disease outcomes, we assumed a 5-year lag for cardiovascular diseases and diabetes (in the first year we assumed one-fifth of the effect of BMI on disease risk, in the second year two-fifths, etc., until 5 years, when the full effect is assumed) and a 20-year lag for all cancers. In addition to obesity-related disease, we simulated changes in the burden of all other diseases arising from changes in life expectancy, by making allowance for disutility associated with age that is not attributed to the diseases specifically included in the model.

In addition, we simulated the effect of weight status in childhood on quality of life using utility weights associated with being in the healthy, overweight, and obese BMI ranges [49].

Thus, the total effect on DALYs is the sum of the effect in childhood (from the direct effect of obesity and overweight on quality of life) and the effect in adulthood mediated through the different diseases.

### Costs attributable to changes in disease incidence

We estimated the change in healthcare costs, social care costs, and employment costs associated with the modelled scenarios. The healthcare costs were derived from NHS England programme budgeting data [45]. The costs included both unit costs for all modelled diseases and unit costs for all other non-modelled diseases, which allowed our modelling to account for potential increases in overall healthcare costs due to increases in life expectancy (often called unrelated future healthcare costs). All costs were adjusted from 2014 to the baseline year of 2015 using the consumer price index for health.

### Health-related net monetary benefit

We estimated the health-related net monetary benefit attributable to changes in health from this policy by summing the cost changes attributable to changes in disease incidence and consequent healthcare cost changes (as described above) and the monetary value attributed to changes in health status. For the latter, was assumed that 1 DALY was equivalent to £60,000, based on the UK Treasury guidance that 1 quality-adjusted life year (QALY) should be valued at £60,000 [47]. DALYs are, roughly speaking, the reciprocal of QALYs. This is also in keeping with the WHO-CHOICE method for valuing a DALY averted (1 to 3 times the GDP per capita; in 2015, GDP per capita in the UK was £28,983, thus giving a mid-point estimate of a DALY of £57,966) [50,51].

### Uncertainty and sensitivity analyses

We determined 95% uncertainty intervals (UIs) for our main outcome measures using Monte Carlo analysis. To do this we ran the model 1,000 times; for each run, we used a different estimate for each parameter, drawn from the distribution described by the parameter's mean and standard deviation.

We undertook a set of sensitivity analyses, where we tested the effect of key assumptions on the model outputs (obesity and DALYs) under scenario A. These analyses are summarised in Table 2.

In addition, we undertook a further set of sensitivity analyses in which we explored the effect of assuming that older children (aged 15–17 years) had the same caloric response to exposure to HFSS adverts as younger children (aged 5–14 years). As we lacked detail on the viewing patterns of these older children, we assumed their viewing patterns were either like those of children aged 4–15 years or like those of people aged 16–65 years (i.e., the 2 age groups on which we had data on viewing behaviours). We ran these sensitivity analyses on both scenarios A and B, to explore a potential concern around older children seeing more HFSS adverts if they watched television after 21.00 hours.

## Results

The descriptive characteristics of the study population are shown in Table 3. On average, children in the UK saw 2.1 HFSS adverts on television per day, of which 71.5% (1.5) were between 05.30 hours and 21.00 hours. If all of these HFSS adverts were withdrawn in response to the restrictions (i.e., scenario A), we estimate that children's mean daily energy intake would decrease by 9.1 kcal (95% UI 0.5–17.7 kcal). This calorie reduction would reduce the number

**Table 2. Description of sensitivity analyses.**

| Sensitivity analysis | Description |
|---|---|
| 2018 nutrient profiling model | We used the revised version of the nutrient profiling model, published in 2018. This model has revised thresholds for sugar and is more liable to classify food with sugar as HFSS [52]. |
| Classification of brand adverts—all HFSS | We assumed all adverts for brands that contained a mix of HFSS and non-HFSS products included HFSS products within the advert and would be subject to the restrictions. |
| Classification of brand adverts—all non-HFSS | We assumed all adverts for brands that contained a mix of HFSS and non-HFSS products included only non-HFSS products within the advert and would not be subject to the restrictions. |
| NICE discount rate | We used a discount rate of 3.5% for costs and DALYs (with no tapering over time), instead of those recommended by the UK Treasury. The 3.5% discount rate is recommended by NICE for evaluation of health and care interventions in the UK. |
| Differential impact by weight | Assuming the same average change in energy intake, we assumed a greater effect of the proposed regulations on children with overweight than on children with normal weight. Based on the Russell et al. [19] meta-analysis, we assumed that children with overweight consumed more calories in response to HFSS advertising than children with normal weight (125.5 kcal versus 79.9 kcal, ratio 1.57); that children with overweight watched more television, and thus were more exposed to HFSS adverts (based on Health Survey for England 2008 data, obese children watch 120 minutes versus 102.9 minutes for those who are not obese; ratio 1.17); and that the cumulative effect of the increased exposure and increased sensitivity to exposure resulted in an approximate 80% difference in change in energy intake ($1.57 \times 1.17 = 1.83$). |

DALY, disability-adjusted life year; HFSS, high in fat, sugar, and salt; NICE, National Institute for Health and Care Excellence.

of children aged 5–17 years with obesity by 4.6% (95% UI 1.4%–9.5%) and with overweight by 3.6% (95% UI 1.1%–7.4%) (Table 4).

This is equivalent to 40,000 (95% UI 12,000–81,000) fewer children with obesity, and 120,000 (95% UI 34,000–240,000) fewer with overweight, based on the 2015 population using the IOTF cut-points. This is also equivalent to a number needed to treat (NNT) of 22 for obesity and 28 for overweight, or reducing the point prevalence of obesity by 0.4 percentage points, from 8.8% to 8.4%, and the point prevalence of overweight by 1.2 percentage points, from 33.1% to 31.9%.

**Table 3. Characteristics of the study population at baseline ($n$ = 13,729,000 children aged 0–17 years in 2015).**

| Characteristic | $n$ (%), $n$, or mean |
|---|---|
| Number of children (5–17 years) with obesity (IOTF) | 852,000 (8.8%) |
| Number of children (5–17 years) with overweight (IOTF) | 3,210,000 (33.1%) |
| Number of children (5–17 years) with obesity (UK90, > 95th centile) | 2,240,000 (23.1%) |
| Number of children (5–17 years) with overweight (UK90, > 85th centile) | 3,890,000 (40.1%) |
| Child impacts (4–15 years) per year for food adverts (millions) | 14,200 |
| Child impacts (4–15 years) per year for HFSS adverts (millions) | 6,640 |
| Child impacts (4–15 years) per year for HFSS adverts between 05.30 and 21.00 hours (millions) | 4,740 |
| Number of food adverts seen per child (4–15 years) per day | 4.45 |
| Number of HFSS adverts seen per child (4–15 years) per day | 2.07 |
| Number of HFSS adverts seen per child (4–15 years) per day between 05.30 and 21.00 hours | 1.48 |

HFSS, high in fat, sugar, and salt; IOTF, International Obesity Task Force.

**Table 4. Estimated impact on children's energy intake and weight status of restricting HFSS advertising on television between 05.30 hours and 21.00 hours.**

| Scenario | Mean reduction in HFSS adverts seen per day | Mean reduction in energy intake (kcal/day) (95% UI) | Reduction in number of children aged 5–17 years with obesity (95% UI) | | Reduction in number of children aged 5–17 years with overweight (95% UI) | |
|---|---|---|---|---|---|---|
| | | | IOTF cut-points | UK90 cut-points | IOTF cut-points | UK90 cut-points |
| A | 1.5 | 9.1 (0.5–17.7) | 40,000 (12,000–81,000); 4.6% (1.4%–9.5%) | 88,000 (15,000–300,000); 3.9% (1.2%–8.8%) | 120,000 (34,000–240,000); 3.6% (1.1%–7.4%) | 130,000 (22,000–440,000); 3.3% (1.0–7.4%) |
| B | 0.5 | 2.8 (0.1–5.3) | 12,000 (3,100–28,000); 1.4% (0.4%–3.3%) | 27,000 (7,400–63,000); 1.2% (0.3%–2.8%) | 35,000 (9,000–81,000); 1.1% (0.3%–2.5%) | 39,000 (11,000–92,000); 1.0% (0.3%–2.4%) |

Estimates of number of children are based on 2015 population. Scenario A: all advertising between 05.30 hours and 21.00 hours is withdrawn. Scenario B: all advertising between 05.30 hours and 21.00 hours is displaced to 21.00 hours to 05.30 hours. UK90 cut-points based on UK90 growth chart, using the 85th and 95th centile.

HFSS, high in fat, sugar, and salt; IOTF, International Obesity Task Force; UI, uncertainty interval.

If the adverts affected by the intervention were shifted to between 21.00 hours and 05.30 hours (scenario B), we estimate the effects would be a quarter to a third of those of scenario A, in which the adverts were withdrawn. Under scenario B, we estimate that the net reduction in HFSS adverts seen by children on average would be 0.5 HFSS adverts per day, and that children's mean daily energy intake would decrease by 2.8 kcal (95% UI 0.1–5.3 kcal), reducing the number of children aged 5–17 years with obesity by 1.4% (95% UI 0.4%–3.3%) and with overweight by 1.1% (95% UI 0.3%–2.5%) (Table 3).

The population attributable fraction of childhood obesity attributable to HFSS advertising (i.e., the proportion of childhood obesity caused by HFSS advertising) was estimated at 6.4% (95% UI 2.0%–13.8%), and of overweight at 5.0% (95% UI 1.5%–10.9%).

We estimated that the intervention would be more effective at reducing obesity and overweight in children from social grade DE than those from social grade AB (2.5-fold greater reduction in number of children with obesity, 6,500 versus 2,600 per million children, and 1.8-fold greater percentage reduction in children with obesity, 6.1% versus 3.5%; 2.1-fold greater reduction in number of children with overweight, 18,000 versus 8,600, and 1.6-fold greater percentage reduction, 4.6% versus 3.0%), as shown in Table 5.

**Table 5. Estimated impact on children's energy intake and weight status of restricting HFSS advertising on television between 05.30 hours and 21.00 hours by social grade.**

| Social grade | Proportion of population | Baseline number of children with obesity and overweight per million (%) | | Reduction in obesity and overweight | | | |
|---|---|---|---|---|---|---|---|
| | | Obesity | Overweight | Obesity | | Overweight | |
| | | | | Number of children per million (95% UI) | Percentage reduction (95% UI) | Number of children per million (95% UI) | Percentage reduction (95% UI) |
| AB | 23% | 75,500 (7.5%) | 294,000 (29.4%) | 2,600 (1,100–4700) | 3.5% (1.4%–6.3%) | 8,600 (3,700–16,000) | 3.0% (1.3%–5.3%) |
| C | 49% | 86,000 (8.6%) | 327,000 (38.4%) | 3,700 (1,300–7,700) | 4.3% (1.5%–9.1%) | 11,000 (3,600–23,500) | 3.4% (1.1%–7.3%) |
| DE | 28% | 107,000 (10.7%) | 384,000 (33.2%) | 6,500 (2,200–14,000) | 6.1% (1.1%–20.6%) | 18,000 (5,500–38,000) | 4.6% (1.4%–10.0%) |

Estimates are based on International Obesity Task Force cut-points. Social grade defined using the 5-point National Readership Survey classification based on occupation of head of household.

HFSS, high in fat, sugar, and salt; UI, uncertainty interval.

**Table 6. Estimated lifetime benefits for today's children (*n* = 13,729,000) attributable to changes in weight status due to restrictions in HFSS advertising on television between 05.30 hours and 21.00 hours.**

| Scenario | Total DALYs (undiscounted) | Total DALYs (discounted) | DALYs in childhood (discounted) | Healthcare cost savings (millions) | Social care cost savings (millions) | Change in production/ employment costs (millions) | Health-related net monetary benefit (millions) |
|---|---|---|---|---|---|---|---|
| A | 240,000 (65,000–530,000) | 120,000 (32,000–250,000) | 37,000 (11,000–90,000) | £84 (£23–£190) | £210 (£56–£490) | £200 (£55–£440) | £7,400 (£2,000–£16,000) |
| B | 73,000 (20,000–160,000) | 35,000 (9,700–78,000) | 7,700 (2,300–17,000) | £26 (£7–£57) | £64 (£17–£150) | £61 (£17–£130) | £2,200 (£620–£4,900) |

Values in parentheses are 95% uncertainty intervals. Scenario A: all advertising between 05.30 hours and 21.00 hours is withdrawn. Scenario B: all advertising between 05.30 hours and 21.00 hours is displaced to 21.00 hours to 05.30 hours.

DALY, disability-adjusted life year; HFSS, high in fat, sugar, and salt.

The lifetime benefits for today's children attributable to changes in weight status arising from restricting HFSS advertising on television between 05.30 hours and 21.00 hours are shown in Table 6. Under scenario A, the restrictions would result in averting 240,000 (95% UI 65,000–530,000) DALYs, resulting in a health-related net monetary benefit of £7.4 billion (95% UI £2.0 billion–£16 billion). The benefits would be between a quarter and a third as great under scenario B, averting 73,000 (95% UI 20,000–160,000) DALYs, with a health-related net monetary benefit of £2.2 billion (95% UI £0.62 billion–£4.9 billion). Under both scenarios, around a fifth (22%) of the gain in health, expressed in DALYs (discounted), occurs during childhood.

The health benefits, expressed in terms of reduction in the number of incident cases of ischaemic heart disease, stroke, diabetes, cancer, and cirrhosis, are shown in Table B in S1 Data.

Sensitivity analyses are shown in Table C and Table D in S1 Data. Of note, assuming that children with overweight or obesity are more likely to watch HFSS adverts and are more sensitive to HFSS adverts increases the estimates of the policy's impact on reduction in the number of children with obesity or overweight by nearly a third. Using the revised nutrient profiling model (published in 2018) [52] increases the estimates of the policy's impact in terms of reducing the number of children with obesity or overweight by over a quarter. If older children's television viewing patterns resemble those of adults (rather than those of children), under scenario B (i.e., all adverts shifted to after 21.00 hours), we estimate that they would see more HFSS adverts than if their viewing patterns resemble those of younger children. Under this scenario, the net effect of the policy would still be to reduce childhood overweight and obesity.

## Discussion

We estimate that restricting HFSS advertising to between 21.00 hours and 05.30 hours has the potential to reduce the number of children aged 5–17 years with obesity by 40,000 (4.6%) and with overweight by 120,000 (3.6%). This could result in averting 240,000 DALYs, with a health-related net monetary benefit of £7.4 billion, for today's children across their lifetime. We estimate the reductions in obesity would be approximately 2-fold greater amongst children in the least affluent social grade compared to the most affluent. The reductions in obesity (57,000, 6.7%) and overweight (170,000, 5.2%) are greater if we assume that children with obesity and overweight are more affected by the restrictions. The benefits are likely to be reduced if HFSS adverts previously shown between 05.30 hours and 21.00 hours are displaced to between 21.00 hours and 05.30 hours; in an extreme scenario of 100% advertisement displacement instead of withdrawal, the benefits could be reduced to between a quarter and a third of these estimates.

## Strengths

The key strength of this analysis is that it extends previous work on the impact of less-healthy television food advertising on children's calorie consumption to quantify the impact on overweight, obesity, and health of a proposed policy to restrict HFSS advertising on television in the UK. In this analysis, we have built on the approach used by others [30] and used a validated set of equations describing the relationship between calorie intake, physical activity, and body fat mass [34]. In contrast to the Australian study [30], we used a recent and more comprehensive meta-analysis quantifying the impact of food advertising on calorie consumption, used real and granular data on children's exposure to HFSS advertising, additionally explored the impact by socioeconomic status (SES), and explored the potential impact of industry's response to the policy. While the results may not be generalisable per se, the method for quantifying the impact of HFSS advertising interventions on children's obesity and overweight can be readily generalised to other settings.

Long-term trials or studies that could test the relationship between exposure to less-healthy television food advertising in children (or adults) and weight gain are likely to be impractical and unfeasible ethically. Modelling studies, such as this, will be an important means and possibly the only means to quantify the impact of less-healthy food advertising on obesity.

## Limitations: Advertising data

We have only considered the direct impact of advertising on children's calorie intake. We have not considered the impact that HFSS food advertising has on changing dietary preferences and habits, which may have a meaningful impact on the dietary behaviours of both children and adults and consequently their health. This will have led to an underestimate of the impact of HFSS advertising on both children's and adult's health.

Our estimates are based on short-term trials that have shown an effect on calorie intake but, being of short duration, have not shown an effect on body weight. A feature of these trials is that children are presented with the opportunity to eat either during or after watching television. The extent to which the findings from these trials hold in real-life, and thus the extent to which the results of these studies can reasonably be extrapolated, is unclear. However, children's television viewing occurs at home, where food is likely to be readily available; much of the viewing occurs at or around meal times; and television viewing is associated with snacking [53,54]. In the Australian study, the estimate of the effect of HFSS advertising on calorie intake was based on 3 trials; in our study, the estimate came from a meta-analysis that included 11 trials from a range of settings, including more naturalistic settings (e.g., schools, holiday camps) [19]. We have assumed a linear relationship between cumulative advertising exposure (measured in time) and calorie consumption. We have not explicitly explored other relationships (e.g., assuming non-linear relationships or accounting for frequency of exposure), which may affect the outcomes. We note the heterogeneity within and wide confidence intervals in (the lower bound of which is close to 0) the meta-analysis describing the effect of less-healthy food advertising on calorie consumption that we used [19]. It is possible that heterogeneity in study design and settings may account for some of this heterogeneity.

From the information we had, we were not able to definitively classify all adverts according to whether they showed HFSS foods or not. However, sensitivity analyses suggest the overall impact of this uncertainty was relatively small, compared to other sources of uncertainty.

We have only considered 1 form of advertising media (television), in part reflecting the absence of good data on exposure to advertising in other media or its effects on consumption. Children's television viewing has declined in recent years, and shifted towards digital viewing [55]. Nonetheless, currently television viewing is likely to be an important source of exposure

to video-based advertisements [56]. If the downward tendency in television exposure is associated with a downward tendency in HFSS advertising exposure (i.e., assuming the pattern, frequency, and duration of advertising has not changed) and it were to continue, then our results are likely to have overestimated the impact of television advertising restrictions on children's weight status.

We have assumed that children aged 5 years watch the same amount of television as children aged 14 years. This assumption may have failed to capture important differences in the impact of the policy by age, particularly under a scenario of advertising shifting to after 21.00 hours (scenario B).

Impacts, as a measure of advertising exposure, do not contain information on the distribution of television viewing within a population, and we assumed a uniform distribution (i.e., all children watch the same amount of television). Given that most children (96%) watch television [35], it seems likely that many children will be affected by these measures by a small amount, rather than a few children being affected by a very large amount. In addition, we have explored scenarios under which TV viewing is not uniform, but varies by social grade and weight status.

## Limitations: Scenarios

The response of marketers to the policy may be complicated, multifaceted, and difficult to estimate before implementation. Scenario B involves *all* HFSS television adverts shifting to after 21.00 hours. This is an extreme scenario of maximum advert shift, and we do not know to what extent that is feasible (e.g., given the number of and cost of available slots after 21.00 hours) or likely (e.g., given the extent to which it duplicates existing advertising). However, other responses are possible: shifting advertising online (although this may not be possible if proposals for a similar watershed online are introduced) [56] or to other media, shifting from product advertising to brand advertising, and increasing use of other marketing strategies (e.g., sponsorship). With our present data, we did not think it was feasible to model these separate responses.

It is also possible that some products might be reformulated in response to any regulations to enable advertising before 21.00 hours to continue, which we have not explored. It is not clear whether this reformulation would lead to an increase or decrease in consumption of HFSS foods relative to scenario A. For example, if advertising of the reformulated products promotes consumption of similarly branded HFSS products, then this would increase consumption of HFSS products. On the other hand, if it encourages people to switch to non-HFSS products, then it would decrease consumption of HFSS products.

## Limitations: Health modelling

We have assumed that the relative change in mean body mass index associated with change in exposure to HFSS television advertising is maintained through life, i.e., if mean BMI is reduced by 0.1 kg/m$^2$ at age 14 years, when this cohort reaches the age of, e.g., 40 years, the mean BMI is reduced by 0.1 kg/m$^2$ compared to the counterfactual. It is unclear to what extent this would be true, although we note weight status tends to track from childhood to adult life, and that populations with higher prevalence of obesity in childhood tend to have a higher prevalence of obesity in adult life [37,38]. Despite this, we note that a significant benefit (around 20% of DALYs averted), albeit discounted benefit, occurs in childhood. So, if one assumed an extreme scenario of no persistence of change in weight status from childhood to adult life, there would still be a significant health benefit (around 20% of that estimated if one assumed 100% persistence of the effect into adult life).

While we have captured many of the important associations between obesity and health, not all associations have been captured. Of note, we have not quantified the impact on musculoskeletal conditions, which are a major cause of morbidity. We have not directly captured the burden of obesity in childhood (e.g., musculoskeletal conditions, asthma, primary care attendance) [4,57–59], although some or much of that may be captured in our quality of life measures. These factors may have led us to underestimate the impact on quality of life and health-related net monetary benefit. We have not included non-disease-related quality of life losses due to overweight and obesity in adulthood, which would have led us to underestimate the health-related net monetary benefit.

We have modelled impacts over the very long term (i.e., following children throughout their life). Over this time period there are many uncertainties that cannot be accounted for, e.g., changes in healthcare technology affecting disease incidence, disease morbidity, and treatment costs, and changes in other risk factors that may affect the underlying incidence of some of the diseases included in the model.

Our primary estimate does not make allowance for differential effects by weight status, which our sensitivity analyses show led to an underestimate in the reductions of the number of children with obesity and overweight by around a third, although the impact of this assumption on population health, measured in DALYs, is relatively small.

Whilst we did estimate differences in BMI and obesity by social grade (a measure of SES), we did not formally estimate the differences in disease by SES as we do not have robust estimates of disease parameters (e.g., incidence and case fatality) by SES. Our estimates of differences in BMI and obesity by SES did not account for possible differences in response by SES (e.g., due to differences in weight status) to television advertising, so may have underestimated the impact of advertising restrictions on health inequalities.

## Limitations: Cost data

We have not considered the costs of the proposal to other sectors of the economy, such as the food industry (e.g., from lost revenue), advertising industry, or commercial television. This is in part because some of the data that would enable us to make these estimates are not publicly available and in part because some of the money not spent on advertising because of restrictions will be spent elsewhere in the economy. For example, if money is not spent on television advertising it may be spent on advertising on different media or returned to investors as profit or customers as reduced prices. Regulatory costs have not been included, as we understand the marginal costs of regulation will be small or even negligible; there is an existing regulatory structure (the costs of which are borne by industry) to enforce the current restrictions around children's television that is likely to be used to enforce any new restrictions.

## Comparison with other studies

Our estimates of the impact on mean body mass index are more conservative (0.12 versus 0.35 kg/m$^2$) than the estimates for the impact of restrictions before 2130 in Australia [30]. The difference is predominantly due to greater exposure of Australian children to less-healthy food advertising on television (see Table E in S1 Data), although exposure to advertising in the Australian study was not estimated from empirical data (in contrast to our study). Other studies have also tried to quantify the effect of advertising restrictions on children's mean body mass index and subsequent health measured in health-adjusted life years (HALYs). Comparisons with respect to mean body mass index are probably most informative as they are not confounded by population size, although differences in exposure to advertising and the proposed restrictions will also affect the validity of the comparisons. Many of these other studies are

methodologically weaker, being based on single trials [60,61], cross-sectional data [62], or consensus opinion from Delphi studies [63]. Our estimate is similar to that of some studies [60,64], more conservative than some [62,63], and less conservative than one [61].

## Policy implications

Modelling such as this involves a number of assumptions. It is particularly difficult to understand how the HFSS advertising landscape will adapt in both the short and long term. For this reason, we suggest the modelled scenarios are best understood as explorations of what the policy could achieve rather than predictions of what will happen, with the 2 scenarios capturing the range of responses that might be expected: at one extreme no industry adaptation and at the other maximum shifting of advertising. Our findings suggest that television advertising of less-healthy food, and likely other similar advertising (i.e., video advertising online), makes a meaningful contribution to increasing levels of childhood obesity. Thus, measures that successfully reduce children's exposure to less-healthy food advertising are likely to make a valuable contribution to the government's goal of halving childhood obesity by 2030.

The UK government is currently considering how to implement further HFSS advertising restrictions to protect children. Concerning advertising on television, there are 2 main options under consideration. The first is restrictions between 05.30 hours and 21.00 hours, which would apply only to foods that are currently subject to the government's reformulation or calorie reduction programme [56]. Whilst this is different to what we have modelled, we expect the effects would be very similar. The major foods excluded from the reformulation programmes are plain breads, cereals (e.g., rice, pasta), fruit, vegetables, nuts, oils, cheese, meat and fish. These products are either not classified as HFSS or are not generally advertised. The second option under consideration is restrictions between 05.30 hours and 21.00 hours but with 'advertising freedoms' granted to certain foods that have been reformulated.

Our work shows that HFSS restrictions between 05.30 hours and 21.00 hours may make a meaningful contribution to improving the health of children, reducing childhood overweight and obesity, and reducing inequalities in overweight and obesity in the UK. However, the impact of any policy will depend, in part, on how the advertising and food industries respond to the policy. One reading of our findings might be that scenario B shows that these responses could significantly reduce the policy's effect and that the policy should not be implemented. Another reading might be that the policy needs to be implemented in a form that makes it harder to undermine (e.g., in effect until 2200 or using the revised, more stringent nutrient profiling model published in 2018) and that measures should be put in place to restrict digital and brand advertising. Options that create loopholes, e.g., advertising freedoms or exemptions for certain foods, also risk undermining the policy's effectiveness.

Whilst we have not formally quantified the impact of restrictions on video-based advertising on non-broadcast media (i.e., online), because we did not have data on online exposure, it seems conceivable that similar effects would be observed if the exposure is similar. Broader advertising restrictions (i.e., on television and non-broadcast media, which are both being consulted on) might also be more effective because they would prevent advertising being displaced from one media to another.

## Future research

The difference between the 2 scenarios underscores the uncertainty around how the food and advertising industry will respond to any policy increasing restrictions on television HFSS advertising and highlights the need for a careful evaluation of any new policy to understand its actual impacts. Evaluation needs to be made of the industry response not only in terms of

shifting advertising slots to after 9 PM, but shifting to other media or moving from product advertising to brand advertising. Understanding the viewing patterns of older children (15–17 years), viewing patterns by SES, and the impact of television advertising on older children are gaps in the current literature. Studies quantifying the relationship between food advertising and food purchases for both children and adults are also needed to facilitate modelled estimates of the impact of food advertising on diet quality and its subsequent impact on health.

## Conclusion

Our study demonstrates that less-healthy food advertising on television in the UK is making a meaningful contribution to childhood overweight and obesity overall, as well as contributing to inequalities in it. Measures that have the potential to reduce exposure to less-healthy food advertising, such as the proposal to restrict HFSS advertising between 05.30 hours and 21.00 hours in the UK, could make a meaningful contribution to reducing childhood obesity and overweight and achieving the government's goal of halving childhood obesity by the year 2030. If HFSS advertising is displaced to after 21.00 hours rather than being withdrawn, the impact of the policy will be reduced.

## Supporting information

**S1 Data. Supplementary data.**
(DOCX)

**S1 Text. Study protocol.**
(DOCX)

**S2 Text. Revisions to study protocol.**
(DOCX)

## Acknowledgments

We are grateful to Peter Scarborough for advice on methods development, including the modelling of differential effects by weight status.

## Author Contributions

**Conceptualization:** Oliver T. Mytton, Emma Boyland, Jean Adams, Brendan Collins, Kate Smith, Russell M. Viner, Linda J. Cobiac.

**Data curation:** Oliver T. Mytton, Martin O'Connell, Simon J. Russell, Rebekah Stroud, Linda J. Cobiac.

**Formal analysis:** Oliver T. Mytton, Simon J. Russell, Rebekah Stroud, Linda J. Cobiac.

**Investigation:** Oliver T. Mytton, Jean Adams, Brendan Collins, Simon J. Russell, Kate Smith, Rebekah Stroud, Linda J. Cobiac.

**Methodology:** Oliver T. Mytton, Emma Boyland, Jean Adams, Brendan Collins, Martin O'Connell, Simon J. Russell, Kate Smith, Rebekah Stroud, Russell M. Viner, Linda J. Cobiac.

**Project administration:** Oliver T. Mytton.

**Resources:** Oliver T. Mytton, Russell M. Viner.

**Software:** Linda J. Cobiac.

**Supervision:** Oliver T. Mytton, Emma Boyland, Jean Adams, Brendan Collins, Martin O'Connell, Russell M. Viner, Linda J. Cobiac.

**Validation:** Oliver T. Mytton, Jean Adams, Brendan Collins, Martin O'Connell, Simon J. Russell, Kate Smith, Rebekah Stroud, Russell M. Viner, Linda J. Cobiac.

**Visualization:** Oliver T. Mytton.

**Writing – original draft:** Oliver T. Mytton, Emma Boyland.

**Writing – review & editing:** Oliver T. Mytton, Emma Boyland, Jean Adams, Brendan Collins, Martin O'Connell, Simon J. Russell, Kate Smith, Rebekah Stroud, Russell M. Viner, Linda J. Cobiac.

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
