## [Decision Letter · Decision Letter 0]

6 May 2020

Dear Dr. Mytton,

Thank you very much for submitting your manuscript "Quantifying the potential health impact of restricting less - healthy food and beverage 2 advertising on UK television between 0530 and 2100 : a modelling study" (PMEDICINE-D-19-04447) for consideration at PLOS Medicine. 

Your paper was evaluated by a senior editor and discussed among all the editors here. It was also discussed with an academic editor with relevant expertise, and sent to independent reviewers, including a statistical reviewer. At this stage, please let us know if you would like us to consider your submission for inclusion in the Obesity Special Issue publishing in July 2020. The reviews are appended at the bottom of this email and any accompanying reviewer attachments can be seen via the link below:

[LINK]

In light of these reviews, I am afraid that we will not be able to accept the manuscript for publication in the journal in its current form, but we would like to consider a revised version that addresses the reviewers' and editors' comments. Obviously we cannot make any decision about publication until we have seen the revised manuscript and your response, and we plan to seek re-review by one or more of the reviewers. 

We expect to receive your revised manuscript by May 27 2020 11:59PM. Please email us (plosmedicine@plos.org) if you have any questions or concerns.

We look forward to receiving your revised manuscript. 

Sincerely,

Adya Misra, PhD

Senior Editor 

PLOS Medicine

plosmedicine.org

Title- please ensure the title entered into EM matches the title of your study. I believe the 2 after “beverage” is a typo

Abstract

Please structure your abstract using the PLOS Medicine headings (Background, Methods and Findings, Conclusions). * Please combine the Methods and Findings sections into one section, “Methods and findings”.

Abstract Methods and Findings: * Please ensure that all numbers presented in the abstract are present and identical to numbers presented in the main manuscript text. * Please include the study design, population and setting, number of participants, years during which the study took place, length of follow up, and main outcome measures. * Please quantify the main results (with 95% CIs and p values). * Please include the important dependent variables that are adjusted for in the analyses. * * In the last sentence of the Abstract Methods and Findings section, please describe the main limitation(s) of the study's methodology.

Conclusions- please begin with “our results show” or similar

* Please address the study implications without overreaching what can be concluded from the data; the phrase "In this study, we observed ..." may be useful. * Please interpret the study based on the results presented in the abstract, emphasizing what is new without overstating your conclusions. * Please avoid vague statements such as "these results have major implications for policy/clinical care". Mention only specific implications substantiated by the results. 

Data Availability Statement

The Data Availability Statement (DAS) requires revision. For each data source used in your study: a) If the data are freely or publicly available, note this and state the location of the data: within the paper, in Supporting Information files, or in a public repository (include the DOI or accession number). b) If the data are owned by a third party but freely available upon request, please note this and state the owner of the data set and contact information for data requests (web or email address). Note that a study author cannot be the contact person for the data. c) If the data are not freely available, please describe briefly the ethical, legal, or contractual restriction that prevents you from sharing it. Please also include an appropriate contact (web or email address) for inquiries (again, this cannot be a study author).

Author Summary

References

Please update the bibliography to Vancouver style and place all full stops after square brackets within text. For example xxx [1]. 

Methods

Please remove the section entitled “ethics and public involvement” 

Please briefly provide demographic information for the 5100 homes used in this study

Line 257-258-please report these data or remove this reference

Please ensure that the study is reported according to the STROBE guideline, and include the completed STROBE checklist as Supporting Information. When completing the checklist, please use section and paragraph numbers, rather than page numbers. Please add the following statement, or similar, to the Methods: "This study is reported as per the Strengthening the Reporting of Observational Studies in Epidemiology STROBE guideline (S1 Checklist)." Please report your study according to the relevant guideline, which can be found here: http://www.equator-network.org/

Did your study have a prospective protocol or analysis plan? Please state this (either way) early in the Methods section. a) If a prospective analysis plan (from your funding proposal, IRB or other ethics committee submission, study protocol, or other planning document written before analyzing the data) was used in designing the study, please include the relevant prospectively written document with your revised manuscript as a Supporting Information file to be published alongside your study, and cite it in the Methods section. A legend for this file should be included at the end of your manuscript. b) If no such document exists, please make sure that the Methods section transparently describes when analyses were planned, and when/why any data-driven changes to analyses took place. c) In either case, changes in the analysis-- including those made in response to peer review comments-- should be identified as such in the Methods section of the paper, with rationale

Comments from the reviewers:

Reviewer #1: This paper aims to estimate the health impact of prohibiting the advertising of food and beverages high fat sugar and salt (HFSS) from 0530 to 2100 on television in the UK. This is valuable research in light to reducing childhood obesity and overweight by 2030. Although methodology is complex it is well explained and accurately described. This paper will be of particular interest to those that participate in general strategies for preventing obesity implemented by any government.

Minor issues

Abstract

Abstract does not follow the author's guideline: Background + Methods and Findings + Conclusions

Major issues

Methodology

Methodology is complex but it is well explained and accurately described. Figure 1 (line 107) helps to understand the whole process. 

Television audience data is from 2015 (line 138). We understand the difficulty in finding current open data, but this fact damages the quality of the research and the final conclusions. Companies like Nielsen are used to transfer their data for free for university research, authors should ask for it. It should be noted that television consumption by children in the UK was 97% in 2015 but fell to 72% in 2018, according to Ofcom (Children and Parents: Media Use and Attitudes Report 2018, p. 28). Multi-screen consumption has not been taken into account, it occurs when watching television using the smartphone or tablet at the same time. The impacts that come from digital media are impossible to control and it would be an important limitation. Ofcom's study states that 37% of children aged 5-15 use smartphones. 

Advertising exposure has been measured by impacts (line 140). It would be advisable to have information about the audience in thousands of individuals who have been impacted. It is important to distinguish between impacting a large percentage of these children a few times or impacting the same group many times. It is an important fact to calculate the decrease in childhood obesity.

On line 212 it is said that if television minutes are reduced, the calories consumed will be reduced. Do you have any data that shows that a 20 second ad is less effective than a 30? Sometimes even 10 second ads are more effective because they are based on specific promotions that are very successful among children. Possibly the number of ads is more important than the exposure time.

It is assumed that television consumption by social class follows the same guidelines as in 2009 (line 242). Some new research associate lowest social class with a higher consumption of internet and digital devices by children. This information should be updated.

Overall I think this is a worthwhile study and I hope that the authors will update audience data to improve the research. 

I am looking forward to reading the final version of the manuscript.

Competing Interests: No competing interests were disclosed.

I have read this submission. I believe that I have an appropriate level of expertise to confirm that it is of an acceptable scientific standard.

Reviewer #2: Referee report on "Quantifying the potential health impact of restricting less-healthy food and beverage advertising on UK television between between 0530 and 2100: a modelling study (PMEDICINE-D-19-04447)

The paper provides an assessment on the potential impact of restricting advertising of less healthy food on UK television. A key novelty is the use of advertising data with information on timing and contents of adverts. The paper does not provide new empirical estimates of the effects of the policy on behaviour (and other subsequent outcomes), but combines the above information on advertising with estimates from previous literature on (i) the effect of advertising exposure on calorie intake by children; (ii) the associated weight gain; as well as related (iii) health effects and (iv) effects on healthcare costs. Due to uncertainty regarding producer and advertiser reactions to stricter regulation, the paper compares the effects of two different scenarios, one where all adverts within the specified time frame are withdrawn, and one where all adverts are displaced to slots outside the specified time frame. 

I am sympathetic to the topic of the paper, and view efforts of reducing childhood obesity to be of major societal importance. The paper is clearly written and well-organized. My main questions relate to whether the value added vis-à-vis earlier literature, and the new information generated by the paper are sufficient to warrant publication in PLOS Medicine. Below, I provide some comments and questions:

Main comments:

1) Contribution of the paper. Is the value added / contribution of the paper sufficient for this journal, compared e.g. to the previous Australian study on the same topic? a key argument made by the authors is that advertising and dietary habits of UK children may differ from those found in Australia; but, given this argument, is this just another country specific study, and as such is the contribution sufficient for this journal? Coming from a different discipline, I am ill-equipped to make this judgement, but nevertheless think that this question should be raised. 

The use of genuine and detailed data on advertising is a definite strong point, but is using such data a sufficient novelty. The paper argues that exploring the potential impact of industry's response to the policy, and analyzing effects by SES, are additional contributions. These do not appear to be a key focus however. The potential industry response is only taken into account through comparing the two hypothetical scenarios; the result of the comparison is quite obvious (displacement leading to smaller effects than complete withdrawal). Examining effects by SES is certainly of interest, but necessarily subject to a lot of uncertainty and simplifying assumptions. The point estimates for the different outcomes (table 4) differ slightly (in an intuitive way), but the confidence intervals are very wide and largely overlapping, so not much can be said / learnt in the end about differences across SES. To put it bluntly, we don't really learn much regarding differences between SES (we cannot say with any confidence whether there are any differences, and how large they might be).

2) Taking into account uncertainty. I would like to see more of a discussion on how uncertainty inherent in combining estimates (that are themselves subject to large uncertainty) from various different sources is taken into account. Do you account for uncertainty inherent in all the steps (i) - (iv) above, each involving externally estimated parameters, and how? Already the CI in the first step - effect of an extra minute of advertising exposure on calorie intake - is very large (0.7 to 27.7 kcal per day), and similar uncertainty is likely in the other steps. More details should be given here, to understand whether the CI reported are correct / what assumptions are made, and how they should be interpreted.

Other comments:

3) Related to point #1, the paper refers to differences between tv viewing patterns and advertising practices between UK and Australia. Is it possible to briefly summarize what is known about these differences. 

4) There is a slight contradiction in the text on lines 143-144 "two impacts can represent two different people viewing the same advert once, or one person viewing two different adverts" vs. lines 177-178 "two impacts could refer to two different people viewing an advert once, or one person viewing an advert twice". This is probably just a typo, but should be clarified; viewing the same advert twice probably has a different impact than viewing two different adverts. 

5) Can we think of scenario A as providing an upper bound for the effects of the policy, while scenario B provides a lower bound? if so, it would be helpful to mention this. 

6) Persistence of weight effects. The paper writes that "we assumed the shift in the BMI curve that we observed at age 15 years persisted into later life". This is explained only on p. 16, where you say that you assume that exactly the same absolute change in BMI persists for the entire life of the person. This should be clarified earlier. Also, is it reasonable (there's certainly a correlation in BMI over lifetime, but is this particular assumption justified, and how important is it for the results)?

7) Assumed lag of health effects. This was unclear to me: "Given uncertainty about the immediacy of the effect of change in BMI on disease outcomes, we conservatively assumed a five-year lag for cardiovascular disease and diabetes and a twenty-year lag for all cancers". Given also the assumption about persistence of weight effects, what does this mean? if a child is exposed to advertising e.g. at 10 years of age, or throughout childhood between 4 - 15 years of age, what is assumed about the occurrence of health effects? 

8) "In addition to obesity-related disease, we simulated the increased burden of all other diseases in added years of life" - this is quite vague, what exactly is done here? 

Typos:

- p. 3, line 90: "health" should be "health impacts"

- p. 4, line 108: "effects" should be "affects"

- p. 7, line 185: "is" should be "are"

Reviewer #3: This is a superbly well written and thorough manuscript.

The study design, data sources and methods used are concisely and transparently described.

The statistical techniques and mathematical modelling applied seem to be fit for purpose.

The distribution and statistical assumptions are clearly stated and appear to be reasonably applied.

The authors have performed various sensitivity analyses for key model assumptions, and have accounted for and communicated uncertainty in their results.

The tables and figures are understandable and informative.

The results and conclusions are presented and discussed fairly, and accurately represent the findings of the model.

The limitations are discussed in detail.

All queries and comments I made during my statistical review were subsequently and adequately answered later on in the manuscript, albeit some being raised in the limitations section or covered in the supplementary document.

[LINK]

---

## [Editor Report · Decision Letter 1]

2 Jun 2020

Dear Dr. Mytton,

Thank you very much for re-submitting your manuscript "Quantifying the potential health impact of restricting less - healthy food and beverage advertising on UK television between 0530 and 2100 : a modelling study" (PMEDICINE-D-19-04447R1) for review by PLOS Medicine.

I have discussed the paper with my colleagues and the academic editor and it was also seen again by reviewers. I am pleased to say that provided the remaining editorial and production issues are dealt with we are planning to accept the paper for publication in the journal.

[LINK]

We look forward to receiving the revised manuscript by Jun 09 2020 11:59PM. 

Sincerely,

Adya Misra, PhD

Senior Editor 

PLOS Medicine

plosmedicine.org

Requests from Editors:

Title-we suggest quantifying is removed from the title. We also recommend the hours are changed to say “prime time TV”. For instance “The potential health impact of restricting less-healthy food and beverage advertising during prime-time TV in the UK: a modelling study”. 

The information about data availability only needs to be within the article metadata under the section “Data Availability”. Please include a link for Kantar data and other data sets. At lines 673 “Data sharing: Much of the data we have used is publicly available. The AC Nielsen and the Broadcasters 674 Audience Research Board (BARB) data are commercially available”. Please be specific, instead of ‘much of’ and ensure links are given for BARB as well as Kantar and add to the data statement in EM.

Space is needed between the first square bracket in refs and the last letter of the preceding word.

In the main text (throughout) could you say for example 21.00 hours, instead of just 2100 for clarity

Abstract

In the abstract “net monetary benefit of £7.4 billion” – please clarify for whom this benefit is significant

Limitations must be specific in the abstract, perhaps something like around line 460 

Conclusions- please temper the language about reduction in policy impact if advertisements are moved to after 2100. I suggest using “we estimate” or similar. Also please specify who is impacted. 

Author summary- much of this is written similarly to the abstract. I would suggest rephrasing lines 66-74 using simpler language 

I also recommend lines 76-77 are moved to the third bullet point

Please change “social grade” to “socioeconomic status” or similar

Please provide a table for the social grades rather than mention in text as it would be easier then for referring back to it as reading through the paper? Lines 281-287

Line 603 – please change to ‘may’ have the potential

Please remove text from 640 – 670 or so as this pulls in from EM automatically. 

Comments from Reviewers:

[LINK]

---

## [Editor Report · Decision Letter 2]

8 Sep 2020

Dear Dr Mytton, 

On behalf of my colleagues and the academic editor, Dr. Sanjay Basu, I am delighted to inform you that your manuscript entitled "The potential health impact of restricting less-healthy food and beverage advertising on UK television between 05.30 and 21.00 hours: a modelling study" (PMEDICINE-D-19-04447R2) has been accepted for publication in PLOS Medicine. 

PRODUCTION PROCESS

PRESS

PROFILE INFORMATION

Thank you again for submitting the manuscript to PLOS Medicine. We look forward to publishing it. 

Best wishes, 

Adya Misra, PhD

Senior Editor 

PLOS Medicine

plosmedicine.org